# A Characterization System for the Monitoring of ELI-NP Gamma Beam †

Rita Borgheresi [1,2,*], Oscar Adriani [1,2], Sebastiano Albergo [3,4], Mirco Andreotti [5], Gigi Cappello [3], Paolo Cardarelli [5,6], Roberto Ciaranfi [1], Elisabetta Maria Grazia Consoli [5], Giovanni Di Domenico [5], Federico Evangelisti [5], Mauro Gambaccini [5,6], Giacomo Graziani [1], Michela Lenzi [1], Fernando Maletta [1], Michele Marziani [5], Giovanni Passaleva [1], Gianfranco Paternò [5], Alin Serban [1,‡], Stefano Squerzanti [5], Oleksandr Starodubtsev [1], Alessia Tricomi [3,4], Matteo Turisini [1], Alessandro Variola [7], Michele Veltri [1,8] and Bruno Zerbo [3]

1   Istituto Nazionale di Fisica Nucleare, INFN-Sezione di Firenze, 50019 Firenze, Italy
2   Università degli Studi di Firenze, 50019 Firenze, Italy
3   Istituto Nazionale di Fisica Nucleare, INFN-Sezione di Catania, 95125 Catania, Italy
4   Università degli Studi di Catania, 95123, Catania Italy
5   Istituto Nazionale di Fisica Nucleare, INFN-Sezione di Ferrara, 44122 Ferrara, Italy
6   Università degli Studi di Ferrara, 44122 Ferrara, Italy
7   Istituto Nazionale di Fisica Nucleare, INFN-Laboratori Nazionali di Frascati, 00044 Frascati, Italy
8   Università degli Studi di Urbino, 61029 Urbino, Italy
*   Correspondence: borgheresi@fi.infn.it
†   Presented at the 7th International Conference on New Frontiers in Physics (ICNFP 2018), Crete, Greece, 4–12 July 2018.
‡   On leave from National Institute for Nuclear Physics and Engineering-Horia Hulubei, Magurele, Romania.

**Abstract:** The ELI-NP (Extreme Light Infrastructure-Nuclear Physics) facility, currently under construction near Bucharest (Romania), is the pillar of the project ELI dedicated to the generation of high-brilliance gamma beams and high-power laser pulses that will be used for frontier research in nuclear physics. To develop an experimental program at the frontiers of the present-day knowledge, two pieces of equipment will be deployed at ELI-NP: a high power laser system consisting of two 10 PW lasers and a high brilliance gamma beam system. The ELI-NP Gamma beam system will deliver an intense gamma beam with unprecedented specifications in terms of photon flux, brilliance and energy bandwidth in an energy range from 0.2 to 20 MeV. Such a gamma beam requires special devices and techniques to measure and monitor the beam parameters during the commissioning and the operational phase. To accomplish this task, the Gamma Beam Characterization System, equipped with four elements, was developed: a Compton spectrometer (CSPEC), to measure and monitor the photon energy spectrum; a nuclear resonant scattering system (NRSS), for absolute beam energy calibration and inter-calibration of the other detectors; a beam profile imager (GPI) to be used for alignment and diagnostics purposes; and finally a sampling calorimeter (GCAL), for a fast combined measurement of the beam average energy and intensity. The combination of the measurements performed by GCAL and CSPEC allows fully characterizing the gamma beam energy distribution and intensity with a precision at the level of few per mill, enough to demonstrate the fulfillment of the required parameters. This article presents an overview of the gamma beam characterization system with focus on these two detectors, which were designed, assembled and are currently under test at INFN-Firenze. The layout and the working principle of the four devices is described, as well as some of the main results of detector tests.

**Keywords:** X- and $\gamma$-ray spectroscopy; beam characteristics; calorimeters; scintillation detectors; solid-state detectors

## 1. Introduction

The ELI-NP facility is the pillar of the project ELI dedicated to the generation of high intensity gamma beams for frontier research in nuclear physics. To develop an experimental program at the frontiers of the present-day knowledge, a high brilliance Gamma Beam System (GBS) with the key parameters shown in Table 1 will be deployed at ELI-NP.

The GBS will provide much more intense and smaller bandwidth $\gamma$ beams (flux $10^{13}\gamma/s$, bandwidth 0.5%) than the presently more powerful $\gamma$-beam facilities, e.g. HI$\gamma$S at Duke University (USA, [1], flux $10^8\gamma/s$, bandwidth 5%), or NewSUBARU at SPring-8 (Japan, [2], flux $10^5 - 10^6\gamma/s$, bandwidth $1 - 2\%$).

**Table 1.** Summary of gamma-ray beam specifications [3].

| Parameter | Value |
|---|---|
| Photon energy | 0.2–19.5 MeV |
| Bandwidth | $\leq$0.5% |
| Spectral density | $(0.8–4)10^4$ ph/s/eV |
| # photons per pulse within FWHM bdw. | $\leq 2.6 \times 10^5$ |
| Number of pulses/macro-pulse | 32 |
| Source rms size | 10–30 μm |
| Source rms divergence | 25–200 μrad |
| Peak brilliance | $10^{20}$–$10^{23}$ $\left(\frac{1}{\text{sec·mm}^2\text{mrad}^2 0.1\%\text{BW}}\right)$ |
| Linear polarization | >99% |
| Energy jitter pulse-to-pulse | <0.2% |
| # photons jitter pulse-to-pulse | $\leq 3\,\%$ |
| Macro repetition rate | 100 Hz |
| Pulse-to-pulse separation | 16 ns |

The GBS will be obtained by collimating the radiation emerging from inverse Compton scattering of short laser pulses on relativistic electron beam bunches. To cover the entire energy range, the GBS will consist of two parallel beamlines, one for gamma energies ranging from 0.2 to 3.5 MeV, and the other from 3.5 to 19.5 after a further acceleration of the electron beam.

A precise energy calibration of the gamma beam and the monitoring of the stability of its parameters, as well as a fast feedback on the energy distribution, intensity and shape profile of the beam, are essential for the commissioning and development of the source. Furthermore, during standard operation, the ELI-NP-GBS will need a monitoring system of these parameters for routine diagnostic of the delivered beam. For these reasons, a dedicated detector system, named Gamma Beam Characterization System [3], was designed. To provide a continuous monitor of the gamma beam energy, without interference with physics data taking, instead of using a diagnostic system directly placed on the beam path with absorbers to reduce its intensity (such as the one used in HI$\gamma$S facility [4]), an approach based on Compton scattering of the beam on a light target was investigated. Typically, the scattered photons are detected in a HPGe detector [5,6] with a resolution on the gamma energy limited by the geometry of the detection system to few percent. Furthermore, at higher energies, the full energy peak is not easily distinguishable from the Compton background and unfolding techniques must be applied. Instead, as described in Section 2, we decided to detect, and precisely track, the scattered electrons (see also [7] and [8]).

For each energy beamline, a complete collimation and gamma beam characterization systems will be realized. The two systems consist of four detectors (see Figure 1): a Compton spectrometer (CSPEC), to measure and monitor the photon energy spectrum; a nuclear resonant scattering system (NRSS), for absolute beam energy calibration and inter-calibration of the other detectors; a beam profile imager (GPI) to be used for alignment and diagnostics purposes; and, finally, a sampling calorimeter (GCAL), for a fast combined measurement of the beam average energy and intensity.

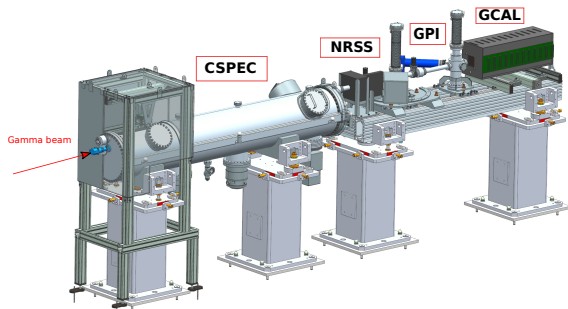

**Figure 1.** Overview of the Gamma Beam Characterization System and of its four detectors. The detectors are labeled with the acronyms presented in the text.

In this work, we describe the design and the tests carried out on the detectors, which have been realized for the characterization of the low energy beamline (LE). In fact, the low energy line is the first that will be realized.

## 2. Compton Spectrometer

The aim of the Compton spectrometer is to reconstruct the energy spectrum of the $\gamma$ beam with a non-destructive method, suitable for both beam characterization during the commissioning phase and beam monitoring during routine operations of the ELI-NP facility.

The basic idea is to measure the energy and position of electrons recoiling at small angles from Compton interactions of the $\gamma$ beam on a thin Mylar target with thickness ranging from 2 to 100 μm.

Two meters downstream of the interaction point, a high purity germanium detector (HPGe), positioned at 60 mrad below the beam line, with an angular acceptance of 13 mrad, will be used to precisely measure the electron kinetic energy. The HPGe crystal is built in planar configuration by CANBERRA and has a cylindrical shape with a diameter of 80 mm and a thickness of 20 mm. Placed in front of the HPGe detector, a double sided silicon strip detector will determine the impact point of the $e^-$ providing the electron scattering angle $\theta$ with a precision better than 1 mrad. The silicon strip detector has dimensions of $5.33 \times 7$ cm$^2$ and 300 μm thickness. On each side of the sensor, 1024 strips, implanted along orthogonal directions, are readout by 8 VA1 chips [9].

The recoil photon is detected outside the vacuum by Barium Fluoride (BaF$_2$) crystals, whose fast response in coincidence with the HPGe signal will provide the trigger of the system. Monte Carlo (MC) simulations, done using GEANT4 software package [10], show how this coincidence will be very effective in suppressing the background. The request of a hit in the Si strip strongly reduces the events due to a Compton photon, while the detection of a recoil gamma-ray in coincidence suppresses the background due to pair production inside the target. For all the simulated energies, more than 99% of the selected events contain an electron generated by a Compton interaction in the target (see [3] for details). The BaF$_2$ crystals are arranged in a small calorimeter of $4 \times 4$ crystals. Their size is $1.2 \times 1.2 \times 5$ cm$^3$ and they are read out by a multianode PMT manufactured by HAMAMATSU (H12700A-3 model). Signals are acquired with a CAEN V1742 switched capacitor digitizer working at a sampling rate of 1 GS/s.

The CSPEC is expected to reconstruct the $\gamma$ beam energy spectrum with a relative uncertainty, from about 0.3% at 1 MeV down to 0.1% at 3 MeV for the beam peak energy. The estimated relative uncertainty on the beam width, after deconvolution from the experimental resolution, is 0.2%÷0.4% in the same energy range.

The main components and the working principle of the spectrometer are illustrated in Figure 2.

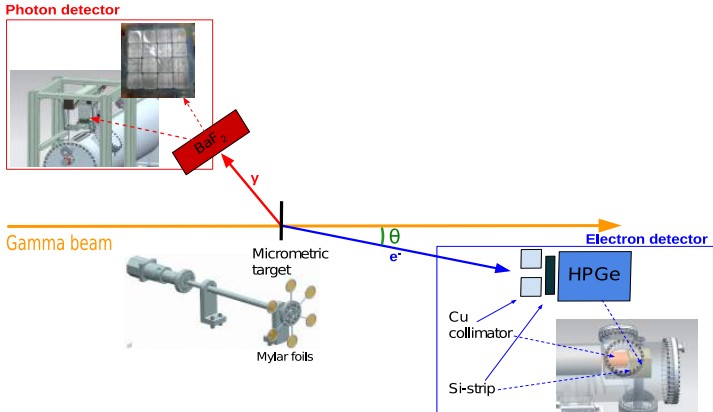

**Figure 2.** Schematic view of the main components of the Compton spectrometer described in the text.

*2.1. Result of Detectors Tests*

2.1.1. HPGe Detector Tests

The resolution on the beam energy measurement critically depends on the accuracy of the electron energy determination, which is correlated to the HPGe energy resolution and to the spread of the energy lost in the materials preceding the HPGe active volume, namely a Be window of 100 μm thickness and the detector's contact approximately 1 μm thick. To partially compensate this loss, the average values, obtained with MC simulations, will be added to the energy measured by the HPGe.

We verified the excellent energy resolution [11] and linearity of the HPGe exposing the detector to different radioactive $\gamma$ sources and obtained a resolution of 0.156% at 1332 keV.

In addition, the accuracy of the HPGe MC simulations, in particular the thickness of the dead layers used in the geometry description of the detector, was verified with a dedicated measurement in vacuum using the conversion electrons of definite energy emitted by a $^{207}$Bi source. To verify the agreement between the measurement and the simulation, a fit procedure has been implemented. The used fit function is the sum of a Crystal Ball, a function with a Gaussian core portion and a power-law low-end tail, used to fit the peak, with an exponential one, for the background. Table 2 reports the fit results for both the measured and the simulated spectra. The measured peak positions, as expected, are lower than the nominal energy of the conversion electron due to the energy loss in the inactive layers. As can be seen, they are in excellent agreement with the simulates one with a precision better than 1 keV. The MC simulation also describes well the $\sigma$ of the gaussian part of the fit function, for which we measure values that differ less than 0.4 keV from the expected ones.

**Table 2.** This table reports the peak positions ($\mu$) and the $\sigma$ of the Gaussian part of the fit function relative to the conversion electron peaks of a $^{207}$Bi source measured (left column) and expected from the MC simulation (right column). The first column reports the nominal conversion electron energies [12].

| $e^-$ peak | DATA | | MC | |
|---|---|---|---|---|
| [keV] | $\mu$ [keV] | $\sigma$ [keV] | $\mu$ [keV] | $\sigma$ [keV] |
| 482 | $452.3 \pm 0.1$ | $4.16 \pm 0.12$ | $452.5 \pm 0.2$ | $4.46 \pm 0.19$ |
| 976 | $951.46 \pm 0.04$ | $2.88 \pm 0.03$ | $952.67 \pm 0.03$ | $2.48 \pm 0.02$ |
| 1048 | $1024.1 \pm 0.1$ | $3.55 \pm 0.08$ | $1025.05 \pm 0.07$ | $3.38 \pm 0.07$ |
| 1060 | $1036.6 \pm 0.1$ | $3.0 \pm 0.1$ | $1037.11 \pm 0.07$ | $3.12 \pm 0.08$ |

2.1.2. BaF$_2$ Detector Tests

BaF$_2$ is a material widely used for timing applications due to the presence of a fast scintillation component ($\tau \sim 0.7$ ns, peaked at 220 nm) alongside with a slow one ($\tau \sim 630$ ns). We implemented

a signal shape identification method that uses the ratio between the maximum value of the fast component and the average of the slow component to discriminate between signals produced by $\gamma$ and those due to thermal noise or to $\alpha$ particles. These $\alpha$ particles originate from the $BaF_2$ crystals' intrinsic radioactivity [13] due to radium impurities, which are always present as radium and barium are homologous elements. The spectrum is dominated by the four alpha lines from the decay chain of $^{226}Ra$ in an energy range from about 4.8 MeV to 7.7 MeV.

The light yield of the scintillator, and consequently the energy calibration, usually depends on the impinging particle type. Therefore, the information obtained from the acquisition of the alpha lines of the radium impurities cannot be used directly to calibrate the $BaF_2$ energy spectrum, as needed for gamma measurements. However, using the alpha peak position as internal standard, a change in the detector performances can be inferred and a correction implemented. This eventual variation might be due to many different effects, for example to a change of the photomultiplier gain or of the coupling between the crystals and the PMT. We plan to use the position of the well-isolated 7.7 MeV line for this purpose, which is shown in Figure 3 (rightmost peak), where the obtained energy spectrum from the $\alpha$ signals is shown as example.

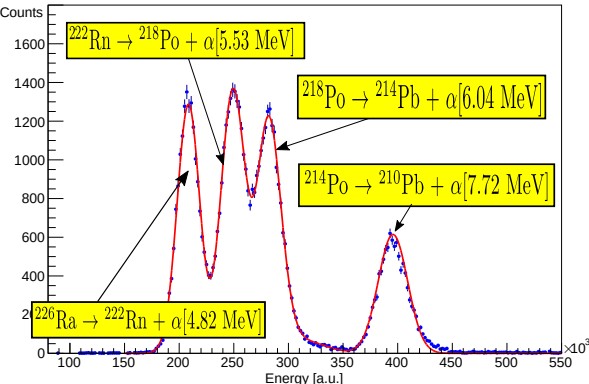

**Figure 3.** Energy spectrum of alpha-particles for one channel. The events were selected using the signal shape identification method. Superimposed as a continuous red line on the plot is the fit function built from $\alpha$ decays of $^{228}Ra$ and $^{226}Ra$ as in [13,14]. The labels indicate the major contributions to the spectrum, which stem from the $^{226}Ra$ decay chain.

## 3. Nuclear Resonant Scattering System

The nuclear resonant scattering system will be used to perform an absolute energy calibration of both CSPEC and GCAL, with an accuracy better than 0.1%, as well as to perform a redundant beam energy measurement.

The basic idea of this device is to detect the gamma decays, of properly selected nuclear excited levels when resonant conditions with the beam energy are achieved. The energy positions $E_r$ of the selected levels have been previously reported with high precision in the literature. By varying the $\gamma$-beam energy, a re-emission of gamma-rays will be generated at the resonant condition. The calibration correspondence is then achieved at $\gamma$-beam energy $E = E_r$, with an uncertainty mainly determined by the step-size in the beam energy scan.

The detection setup was designed to measure nuclear resonance scattering from $\gamma$-beam photons at backward angles (around $\theta = 135°$) with respect to the beam direction, as shown in Figure 4. This angular condition is important in order to reduce the background contribution coming from the photon Compton scattered on the target.

The mechanical design, presented in Figure 4, is mainly composed by three parts, namely the scattering chamber, the target holder and the $\gamma$-detector, which is placed outside the vacuum line.

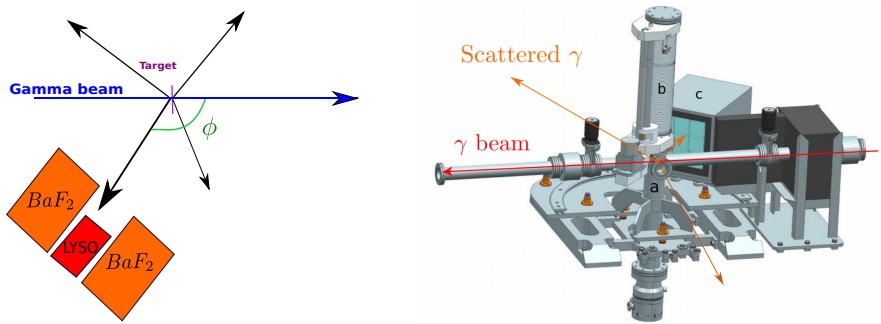

**Figure 4.** Left figure shows a schematic representation of the NRSS. The right one presents the layout of the NRSS: (**a**) the scattering chamber; (**b**) the vertical target shifter; and (**c**) the scintillators crystal for the $\gamma$ detector.

The $\gamma$ detector was designed to work in two different modes: Fast Counter Mode (FC) allows a fast beam energy scan, giving prompt information about the establishment of a resonance condition; and Energy Spectrum Mode (ES) permits the precise identification of the resonant level through the measurement of the energy of the emitted de-excitation photon.

The detector consists of a Lutetium-yttrium oxyorthosilicate (LYSO) crystal of dimensions $3 \times 3 \times 6$ cm$^3$ surrounded by an ensemble of four $5 \times 5 \times 8$ cm$^3$ BaF$_2$ scintillators. These act both as fast counters for the FC mode and as Compton shield for the ES mode, while the LYSO provides the energy of the $\gamma$. The system is placed inside a box with 2 cm-thick lead screen on its side walls.

Detailed background studies qwew performed using a dedicated Geant4 simulation of geometry and realistic beams [15,16]. From these studies emerges that there are two background sources which interfere with the NRSS operations: gamma beam environmental background and target processes competing with the resonant scattering (such as the Compton scattering). The environmental background, in turn, can be distinguished mainly in two different types:

- The first component is due to photons not absorbed by the collimator. They have random directions, due to multiple scatterings on the concrete walls of the room and low energy and enter in the detector from the lateral directions.
- The second component is directly related to the $\gamma$ beam. In fact, these "beam-like" $\gamma$ are due to beam scattering on the edges of the collimators, thus they have the same direction of the beam and a similar energy.

The first component of the environmental background can be rejected with Pb shielding. The second component (the beam-like photons) reaches the NRSS system out of time with respect to the resonant ones and can then be rejected thanks to the excellent time resolution of the system. The most challenging background source is hence generated by the Compton back-scattered photons from the target, which arrive in time with the resonant scattered gamma. At the detection angle of $135°$, this background source emits in an energy region below 500 keV. However, the pile-up arising from the big amount of Compton back-scattered photons makes impossible to rely only on an energy-based rejection. For this reason, a peculiar technique based on dual readout of Cherenkov and scintillation light has been developed. It consists in discrimination between the low Cherenkov emission of pile-up background from the larger one generated by the resonance signals [17]. The power of this method was investigated with the help of MC simulations. Figure 5 shows the expected energy distribution relatives to only the background component and to both the signal and the background components. The simulations reported in Figure 5 refer to a realistic 3 MeV ELI-NP beam, simulated taking into account also collimation effects. The simulated resonance signal refers to the 2.98 MeV level of $^{27}$Al. In the figure, it can be noticed that the background component is highly suppressed using a Cherenkov selection cut, without reducing the signal.

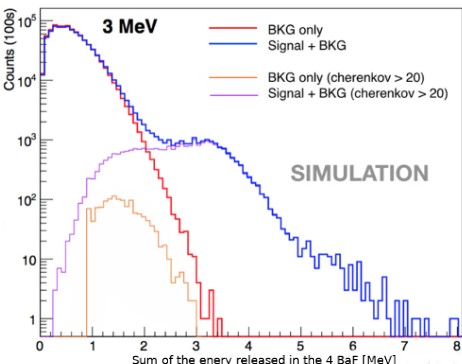

**Figure 5.** Comparison between the recorded energy in BaF$_2$ crystals for a background-only sample (red and orange lines) and for a signal + background sample (blue and violet lines), generated by the interaction of a 3 MeV gamma beam with a 2 mm thick Al target. The thick lines are the distributions obtained without any cut on Cherenkov signals, while the thin ones are obtained with a cut of > 20 Cherenkov photons. Samples refers to 100 s data taking at nominal ELI-NP beam intensity.

## 4. Profile Imager

The task of the gamma beam profile imager is to provide an image of the gamma spatial distribution to display the position of the beam. This image is crucial in giving informations on the alignment of the collimation system and on the correct positioning of the other detectors, as well as to control the size and the shape of the $\gamma$ beam.

The imager is made by a vacuum chamber placed in the beamline and hosting a tilted LYSO scintillator crystal, crossed by the gamma beam at an angle of 45°, and hosted in a target holder, which supports interchangeable targets. Outside the chamber, in air, looking at the target through a quartz viewport, is placed a CCD camera coupled with a lens system to focus the scintillator light. A mirror reflects downwards to the camera the light coming out from the vacuum window. The CCD is mounted on a remotely controlled linear stage for fine focus adjustment. The entire system is enclosed in a dark box to avoid background signals due to environmental light. The adopted solution is shown in Figure 6.

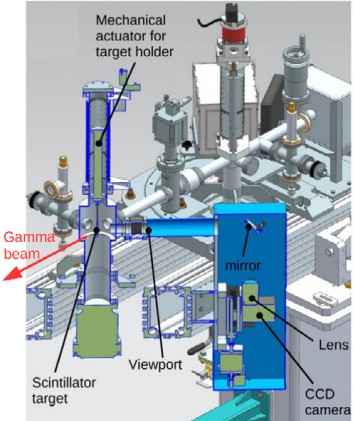
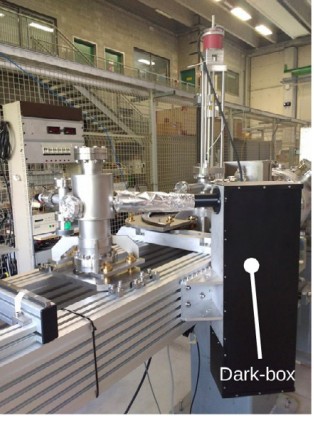

**Figure 6.** On the right is shown the layout of the GPI detector with the display of a cross-section to show the inner parts of the detector, while on the left is displayed the actual assembly of the device at INFN Ferrara.

To predict the imaging system response, an analytical model was developed with the main goal of working out an expression for the signal expected on the CCD as a function of the system configuration. This model was validated by carrying out a set of experimental tests on a prototype by using the photon beam from a Varian M-143T X-ray tube. A full description of the model and the prototype

testing can be found in [18]. By knowing the imaging system response, it was possible to predict the expected signal and spatial resolution with the ELI-NP-GBS beam. The energy deposition inside the LYSO crystal by the $\gamma$ beams of various energies was evaluated through a set of MC simulations using Geant4. The signal on the CCD was then calculated from these data by using the aforementioned analytical model. The expected average signal on CCD images for the ELI-NP-GBS ranges from 305 Gray Level (GL) per second at 0.2 MeV to 51400 GL/s at 19.5 MeV. Therefore, the signal results to be well above the expected noise of about 45 GL for acquisition time of 1 s. The expected signal will allow us to obtain an image of the spatial distribution of the gamma beam in a small amount of time ($\sim$1 s) for the entire energy range.

## 5. Gamma Calorimeter

In a classic nuclear or sub-nuclear experiment, the particles are detected one by one. Indeed, the calorimeter here described must be able to measure the energy of a $\gamma$ beam that is supposed to be approximately monochromatic (see Table 1) but whose intensity is not known. The calorimeter has to be able to record the release of energy due to the simultaneous arrival (within $\sim$1 ps) of a large number of photon all with the same energy. However, the photon energy cannot be simply determined from the total energy released, as usually happens in calorimeters, due to the fact that the intensity is not exactly known. Therefore, the basic idea adopted in GCAL is to use properties of the gamma energy release inside the detector that depend only on the photon energy and not on the beam intensity.

This is obtained using the monotonic energy dependence of the $\gamma$ cross-section for low-Z materials in the energy range of interest at the ELI-NP facility. Indeed, the average depth of a photon interaction inside a light absorber is expected to increase with energy. Therefore, the longitudinal profile of the energy deposition can be used to reconstruct the $\gamma$ beam energy. The analytic expression of the longitudinal profile vs. $E_\gamma$ is not known; therefore, we use MC simulations to parameterize the profile of the energy release as a function of the $\gamma$ beam energy. The reconstruction of the average beam energy ($E_{beam}$) takes place through the comparison of the measured profile against the simulated ones. Once $E_{beam}$ is known, in the assumption of a monochromatic beam, the number of photons in a pulse, $N_\gamma$, can be determined from the total energy $E_{tot}$ measured in the calorimeter:

$$N_\gamma = \frac{E_{tot}}{f(E_{beam}) \cdot E_{beam}} \tag{1}$$

where $f(E_{beam})$ is the fraction of the total energy released in the calorimeter.

To evaluate the performances of the GCAL on the reconstruction of the average energy and intensity of the beam at different energies, the energy reconstruction procedure was tested on an independently generated MC sample. The results in terms of achievable statistical resolution is shown on the left of Figure 7, while the offset performances are presented on the right for the statistics of $10^5$ beam $\gamma$, corresponding to one ELI-NP pulse. As can be seen in the figure, the best possible statistical accuracy for a measurement with a single pulse is a few percent. We note that this uncertainty becomes lower than 0.1% after collecting data corresponding to few seconds of beam operation with its nominal parameters.

Due to the anticorrelation between $N_\gamma$ and $E_{beam}$, which can be noticed from Equation (1), and to the fact that $N_\gamma$ is calculated using the value of $E_{beam}$ obtained from the fit, if $E_{beam}$ is overestimated, then $N_\gamma$ is underestimated and vice versa. This can be observed clearly from the graphs on the right of Figure 7 for $E_{beam}$ lower than 4 MeV. For these reasons, we plan to do a final in-situ calibration, with the help of the NRSS. The resulting corrections are expected to constrain this systematic bias on the energy scale.

Only the statistical errors have been considered in the results presented in Figure 7. The measurement is expected to be limited by systematic uncertainties, notably related to response calibration whose effects are presented in the next section.

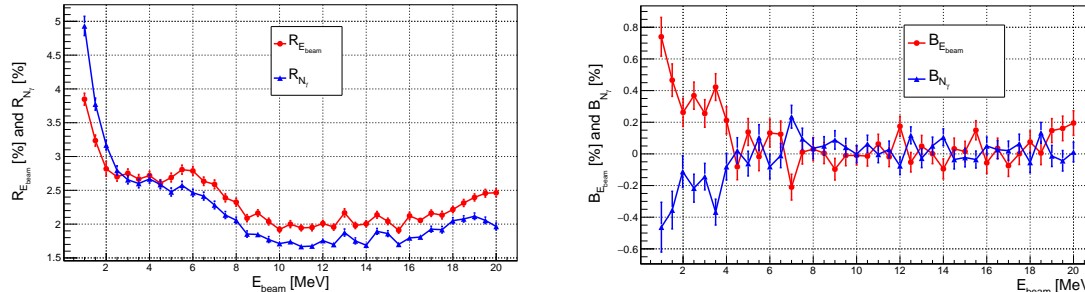

**Figure 7.** (**left**) The expected resolution on beam energy (red dots) and on beam intensity (blue triangles); and (**right**) presents the expected offset on beam energy (red dots) and on beam intensity (blue triangles). Both are relative to a pulse of $10^5$ photons and the plotted quantities are reported a function of the beam energy.

A sampling calorimeter device was chosen to disentangle the requirements of a low atomic number for the absorption material, and of the radiation hardness for the detection technology. The chosen design for the LE beamline is a sampling calorimeter composed by 22 identical layers. Each element consists of a block of polyethylene absorber (an inexpensive and easily workable low-Z material) followed by a readout board hosting seven adjacent silicon detectors. The chosen Si-strip sensors are processed from n-type phosphorous doped wafers, 320 µm thick, segmented in 128 p+ strips. Figure 8 shows on the left a schematic layout of the entire calorimeter and on the right a picture of a single GCAL layer.

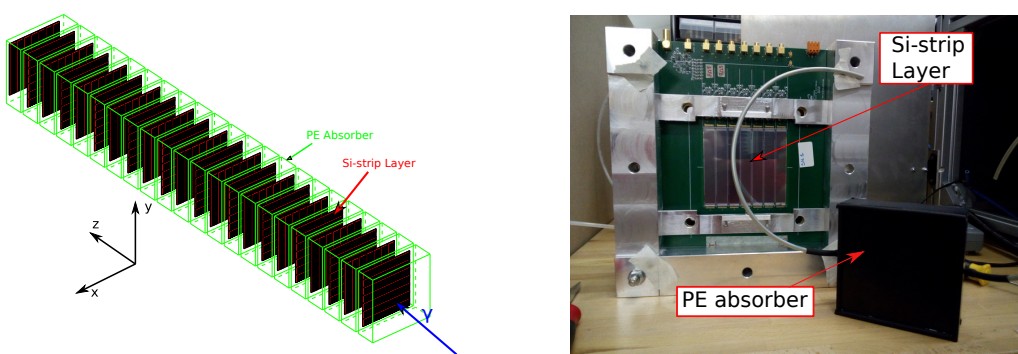

**Figure 8.** Schematic layout of the LE calorimeter (**left**); and a picture of a single GCAL layer (**right**).

*5.1. Silicon Detectors Tests*

The response of the 22 front-end board equipped with the silicon detectors that compose the low-energy line calorimeter were characterized using an infrared laser (IR). In particular, we verified the functionality of all the layers and the detectors time response. Indeed, the time response is a critical issue, since the calorimeter has to be able to resolve the 16 ns separated pulses of the ELI-NP beam.

We used a PicoQuant (model LDH-P-1060) laser diode with variable optical power up to 21 mW and peak wavelength of 1060 nm. To verify that the detectors are able to provide a signal with proper duration even at the highest energy deposition expected for the low-energy beam line, which amounts to about 470 MeV, we adjusted the output laser power in order to obtain this value of energy released inside the detectors. To study the detector response in the most similar condition to the real detector use, the laser was driven with an external trigger in order to obtain a train of 32 pulses with the same temporal structure of the ELI-NP gamma beam. Figure 9 displays the detector response; in particular, the right plot shows an enlargement with the fit function superimposed in red. The fit results were validated using simulated events detecting differences in amplitude of the order of per mill among single pulses. This shows the ability to detect pulses separated by 16 ns with an accuracy at the level

of per mill, allowing the calorimeter to measure the beam energy and intensity and their variation in time within a ELI macro-pulse.

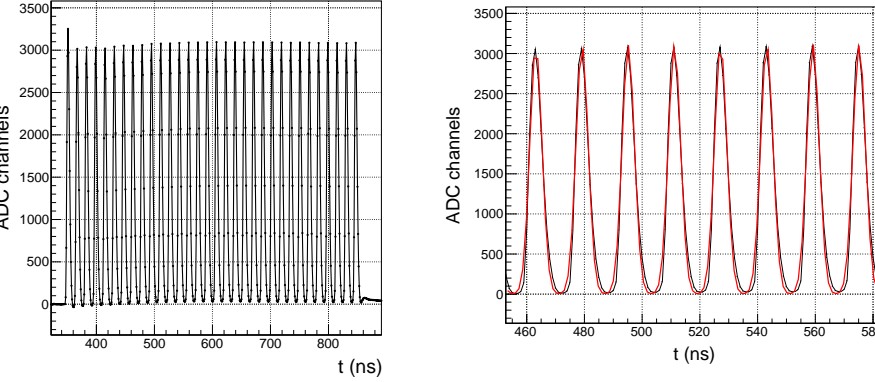

**Figure 9.** In these two graph is shown the train acquisition with 32 pulses input signal. (**right**) An enlargement showing the fit function superimposed in red.

The energy response of the silicon detectors was tested at the LABEC facility in Firenze at the DEFEL [19] beamline using 3 MeV protons. The gains of the calorimeter silicon strip pads were measured. We observed gain differences between sensors belonging to the same board and to different boards randomly distributed according to a Gaussian function with a standard deviation of about 1% [20]. The possible systematic contributions due to the effects of these miscalibrations between different silicon pads were investigated with the help of a toy MC. We found that the presence of this miscalibration effect introduces a systematic shift of the values of the beam energy and intensity, without affecting their resolutions. These additional systematic errors amount to about 0.5% for the energy and 0.7% for the intensity in the low-energy beam range.

*5.2. Conclusion*

A characterization system to measure and monitor the ELI-NP gamma beam parameter has been developed and is under testing. This work presents an update on the status of the four detectors composing the system focusing on the latest results obtained from the GCAL and CSPEC tests. The tests carried out on the CSPEC HPGE detector using electrons of definite energy showing a very good agreement with the simulated ones confirming the correctness of the simulation geometry. The possibility of using the intrinsic $\alpha$ radioactivity of the CSPEC BaF$_2$ detector to monitor eventual changes in the energy calibration was investigated using a selection method to identify the signals due to photons from those due to alpha particle or those due to noise background. Finally, the responses of the calorimeter layers that compose the low-energy calorimeter were characterized showing that the GCAL can disentagle pulses separated by 16 ns.

**Funding:** This research received no external funding.

**Conflicts of Interest:** The authors declare no conflict of interest.

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
