# Peer review of "A Characterization System for the Monitoring of ELI-NP Gamma Beam†"

_proceedings, 2017_

Round 1
Reviewer 1 Report
Report on "A characterization system for the monitoring of ELI-NP gamma beam" by R. Borgheresi et al.
The manuscript describes the characterization system for the quasi-monochromatic gamma beam at the new ELI-NP facility in Bucharest, currently under construction. The authors describe four different separate parts of the system that are in use to characterize the incoming beam in terms of its gamma-energy, areal profile, and intensity.
While I am impressed by the quality of the authors' work developing the beam characterization system, I am sorry to say that this manuscript, in my view, should either undergo a major modification or be withdrawn entirely. I will outline my problems below.
1) There is a large amount of overlap with published work by the same group of authors, some cited in the manuscript, some not even cited. Most are in journals by Elsevier, so in the interest of the journal and publisher of MDPI Universe it must be investigated whether Elsevier will give permission for re-use of the figures. But even if this permission were to be obtained, it would change the character of the manuscript from research article to review, and some necessary elements of a review are completely missing, see below.
I give an incomplete list of the overlap I detected:
* Fig. 1 is already shown in Pellegriti et al. NIM A 865, 60 (2017), with minor changes which do not solve the above mentioned copyright problem. Pellegriti 2017 is not even cited in the manuscript!!
* Fig. 4 right is already shown twice, again with minor updates: Pellegriti op.cit. and Cappello [Ref. 5]
* Fig. 5 seems to be an exact match with Cappello [Ref. 5] Fig. 3 right
* Fig. 6 left = Cardarelli [Ref. 6] Fig. 3b.
* Fig. 6 Cardarelli et al. DOI:10.1016/j.nima.2018.10.049
This overlap must be completely removed, and reference to previous work by the same authors must be fully given.
2) The material in section 2.1.1 and table 2 is completely new, as far as I can tell. It is very interesting but insufficiently documented (for an original research article) or inappropriate (for a review paper). Some questions:
* Is the bismuth source completely open? Some more details are needed to evaluate whether the conversion energy is exactly the nominal one.
* Give proper reference (Nuclear Data Sheets?) and uncertainty also for the nominal conversion electron energies.
* The width of the 1060 keV peak is lower than that of the nearby 1048 keV peak, both in the data and in the simulations. What is the effect causing this? Some physical width of the conversion line? Please discuss.
* The differences between the observed and the nominal energies should be listed, as well. In the simulation, the e- energy loss is consistent at 1048 and 1060 keV, as it should be. In the data, there seems to be significant difference between the two. Please explain.
* For the 482 keV line, the energy loss is almost precisely matched by the simulation, whereas for the higher-energy lines, which should be easier to model, the simulation underpredicts the data by 1.2-0.5 keV. Please explain.
* For the gamma rays used to check the overall linearity [Ref. 2] the whole 2 cm thick HPGe contributes, whereas the electrons are limited to some initial layer of maybe 0.5 cm due to their energy loss. How does this affect the conclusions on the linearity?
* If this section should be published as new original data (I find it sufficiently interesting to justify this!), then it should be expanded and amended along the lines given above and kept separate from any review material.
3) For a review paper, I miss some necessary ingredients:
* Brief scientific justification of the whole ELI-NP gamma beam, detailing why the specs given in Table 1 are needed.
* Same for the beam characterization system shown here.
* What are the required characteristics of the single components of the system, as motivated by the physics goals?
* Overview of all the (seemingly voluminous!) literature produced by the authors, stating the context of the individual studies. It is unacceptable that a simple search like the one I conducted shows uncited, relevant work by the same authors.
* Some necessary general context is missing, e.g. for the Compton apparatus the distances and angles between components, dimensions of collimators, background generated in the collimators, etc.
* There is no reference at all given to other labs working with similar or related techniques. There are a number of gamma beams around the world. ELI-NP will arguably be the best, but hardly the first such facility. Therefore also when diagnostics are discussed, there are previous efforts at other places. They should be mentioned, and the present approach should be compared with them.
4) Fig. 7 is similar to Veltri [Ref. 8] Fig. 2 but shows different structure at 13-14.5 MeV. I found another similar but not equal figure in the TDR [Ref. 1] Fig. 185. What is causing the differences? They seem to be out of the error bars. The changes / progress must be discussed, and the reference to earlier work must be given.
5) Fig. 8 is similar to Veltri [Ref. 8] Fig. 4 but seems to show a different laser shot. It would be instructive to cite, and discuss, the previous work. Actually the comparison between the different laser shots will be helpful to understand which features of the shown pulse trains are just given by shot to shot variations. Instead the earlier work is hidden, and the interested reader is left to find it themselves and wonder what the differences might be.
6) In Figure 3, it is not clear why the Po-210 and Rn-222 lines which lie 0.2 MeV apart are modeled with one single Gaussian while the Po-218 line, which is 0.5 MeV away from the higher of the other lines, is a separate Gaussian. With the resolution given, it should be possible to at least see a hint of a doublet for lines separated by 0.2 MeV. Is one of the two first mentioned isotopes missing? If yes, should they not be in secular equilibrium? Is it a question of branching ratios?
The 7.7 MeV line should in principle be sqrt(7.7/4.9) = 1.25 times wider than the 4.8 MeV line, instead it appears about twice as wide, and it shows some deviations from a Gaussian profile. Please explain.
7) Figure 5, is the signal/noise ratio enough for the low-energy gamma beam application? What is the resultant lower energy limit for the facility?
8) Figure 8, is the useful lower energy limit of the GCAL 1 MeV?
9) Figure 9 is not instructive. It would be better to replace it with a schematic, or at least remove all the cables and tubes so that the structure of the device is visible.
10) The Si detector tests with the laser diode were done with a sinusoidal pulse shape, is this realistic?
11) Any kind of discussion is completely missing. The reader is not even treated to a summary of the work shown.
Summarizing, I have the highest respect for the scientific work undertaken by the authors, but due to the above problems with the manuscript, and the manuscript only, I cannot recommend to publish it.
Author Response
Response provided in the attached file.

Reviewer 2 Report
A characterization system for the monitoring of ELI-NP gamma beam, by Rita Borgheresi et al. (part of the EuroGammaS proposal for the ELI-NP Gamma beam System), submitted to Universe.
The authors describe four different systems they propose to install at the ELI-NP facility to characterize the gamma-beam. These include: “Compton spectrometer (CSPEC), to measure and monitor the photon energy spectrum; a nuclear resonant scattering system (NRSS), for absolute beam energy calibration; a beam profile imager (GPI) to be used for alignment and diagnostics purposes and finally a sampling calorimeter (GCAL), for a fast measurement of the beam average energy and intensity”. The authors are mainly concerned with “fully characterizing the gamma beam energy distribution and intensity… sufficient enough to demonstrate the fulfillment of the required parameters [for the delivery of the gamma-beam system by the EuroGHammaS group, see Ref 1.]”.
The current manuscript does not provide the necessary detailed information of the system to justify the claims. The group does not demonstrate sufficient command of the literature and sufficient knowledge of the state of the art in the field. The proposed system is much inferior to current system(s). The authors failed to communicate with the practitioner in the field (including their “client”, the ELI-NP) and they produce a system that is not “sufficient enough to demonstrate the fulfillment of the required parameters [for the delivery of the gamma-beam system by the EuroGHammaS group]”. For example, the authors failed to join the ELI-NP team in extensive tests to characterize the gamma-beam at the High Intensity gamma Source (HIgS) at Duke University in the USA. Had they joined the test in the USA they would have learned the state of the art in the field and not produce a system that is inferior to the currently available system.
The authors do not demonstrate command of standard nuclear physics techniques for example, for the calibration of HPGe detector with high energy gamma-rays by using a Pu-13C source to produce the 6.129893 MeV +/- 0.04 keV calibration gamma-ray, or the use of the capture on nickel of moderated neutrons from a Pu-Be source to produce a number of gamma-rays with energies between 7.5 and 9.0 MeV [Joel G. Rogers et al., NIMA 413(1998)249].
I comment on each system below:
Compton spectrometer (CSPEC), to measure and monitor the photon energy spectrum:
“The aim of the Compton spectrometer (CSPEC) is to reconstruct the energy spectrum of the gamma-beam” by measuring on-line the energy of the Compton electrons (in coincidence with the Compton gamma-rays). The energy of the Compton electron has a complicated dependence on the energy of the gamma-ray and the angle of the detected electrons. The authors fail to give the necessary details of for example the electron detection angle, angular aperture etc. I have strong doubts of the claimed better than 0.5% energy resolution with which the gamma-beam energy will be reconstructed with the proposed CSPEC system. The required good accuracy of 0.5% demands a detailed discussion of the parameters of the CSPEC which is missing in this paper.
Indeed, if the gamma-beam energy spread (resolution) will be 0.5% as shown in Table 1, it will be best to measure the gamma-beam energy with a resolution which is considerably better than 0.5%. The gamma-beam energy can be easily measured with 0.1-0.2% resolution using current technologies, by placing a HPGe detector in the gamma-beam, as is routinely done at the HIgS gamma-ray facility in the USA [C. Sun et al. NIMA 605(2009)312]. Of course, the HIgS gamma- beam enters the HPGe detector after a large degradation of the intensity using accurately known absorbers.
I note that accurate knowledge of the beam spread (listed 0.5% in Table 1) is immensely important for the experimentalists and it would have been useful to propose a system that is also useful for the experimentalists and not only for “demonstration of delivery by the EuroGHammaS group”.
I fail to see the wisdom of the proposed CSPEC system which is very complicated and inferior to a currently used system which is simple and readily available; i.e. good quality large volume HPGe detectors suitable for measuring high energy gamma-rays are commercially available.
I note that the experience accumulated at the HIgS facility over the last two decades indicate the good stability of the gamma-beam, hence the interruption of the beam by placing a HPGe in the beam path (a few times a day) does not represent a major issue for the research program. An online knowledge of the beam energy may seem nice to have but is not in fact necessary. The offline method used today at the HIgS is preferable due to the higher accuracy.
A nuclear resonant scattering system (NRSS), for absolute beam energy calibration:
This system relies on “energy positions Er of the selected levels [that] have been previously reported with high precision in the literature”. But such data on NRF levels with energies higher than 10 MeV are scarce and the NRF levels are not known with sufficient accuracy. Such a system will be of very limited use due to lack of sufficient NRF resonances with the required accuracy.
A beam profile imager (GPI) to be used for alignment and diagnostics purposes:
A beam profile imager is a standard method which for example was described in the Ph.D. thesis of C. Sun, available from Duke University [Characterizations and Diagnostics of Compton Light Source, C. Sun, Ph.D. thesis, Duke University, 2009]. The authors do not refer to this work and as such do not demonstrate a command of the current literature and state of the art from a decade ago.
A sampling calorimeter (GCAL), for a fast measurement of the beam average energy and intensity:
The propose system “use properties of the gamma energy release inside the detector that depend only on the photon energy and not on the beam intensity”. The authors simulate the energy released by the gamma-beam in the GCAL. But the authors do not discuss the effect of the associated neutrons that are produced by the gamma beam. For example, moderated neutrons that capture on the surrounding material and the GCAL material itself can account for a fraction of the energy deposited in the GCAL. The authors do not consider the effect of neutrons and they fail to give a reliable estimate of the uncertainty in the measured beam average energy and intensity due to the neutrons associated with the gamma-beam.
The authors fail to compare their proposed GCAL to the large (10”) NaI used today at the HIgS facility to measure the beam flux (again, the beam flux is reduced with accurately known absorbers). The commercially available large NaI detector is simple and reliable and it begs the question of why to develop a complicated GCAL when such a simple system can yield the same result. And again the experience accumulated at the HIgS facility over the last two decades indicate the good stability of the gamma-beam, hence the interruption of the beam by placing a large NaI detector in the beam path (a few times a day) does not represent a major issue for the research program.
I do not recommend this paper for publication in “the Universe”. It does not meet elementary requirement of including important necessary details when publishing the proposed system to characterize the gamma-beam. The authors do not demonstrate command of the literature and the authors do not demonstrate knowledge of the state of the art in the field and they propose a complicated system that is inferior to currently used systems that are also considerably simpler.
Author Response
Response provided in the attached file.

Reviewer 3 Report
The paper of Rita Borgheresi et al. “A characterization system for monitoring of ELI-NP gamma beam” reports the gamma-beam characterization system, which has been developed for ELI-NP. The system consists of four elements, which aim at monitoring of different parameters of the gamma beam, e.g. a Compton spectrometer (CSPEC) for measurement and monitoring of the photon energy spectrum, a nuclear resonance scattering system (NRSS) for absolute beam energy calibration, a beam profile imager (GPI) for alignment and diagnostic purposes, and a sampling calorimeter (GCAL) for combined measurement of the beam average energy and intensity.
The paper reads well and describes most of the characteristics of the gamma-beam characterization system. However, a number of issues need to be addressed before this is paper is recommended for publication. First of all, the authors refer only to their own work, six out of eight references. Similar diagnostic systems, which are operational at present, e.g. at the HIγS facility at Duke University and at the NewSUBARU facility at the SPring*8 laboratory are not discussed and it is not possible to compare the designed parameters of the ELI-NP beam diagnostic system with what is operational nowadays. Therefore, I recommend that the authors include such a discussion/comparison in the text. The paper ends abruptly. I think a short summary is needed.
More specific comments:
(1) In the abstract and in the introduction (1st sentence) should be stated that at ELI-NP high-brilliance gamma beams and high-power laser pulses will be used for frontier research in nuclear physics.
(2) I suggest that in Table 1 also the number of pulses per macrobunch is included as a beam parameter.
(3) In Fig. 1 I suggest to indicate the different measurement devices with their abbreviations and in the figure caption to refer to the description in the text.
(4) On page 3, line 69, a reference to the GEANT4 software package is needed, and also “software package” in the text after GEANT4.
(5) The paragraph (lines 67-72) states that the coincidence between the recoil photon, detected by the BaF2 detectors, and the HPGe signal would suppress the background. However, this is only a statement and the authors do not provide any further evidence. I suggest that this issue is discussed in more detail, since the background suppression is important for the functionality of the system.
(6) The inserts of the different systems of the spectrometer, which are included in Fig. 2, need to be described consistently in the text and the figure caption should refer to it. Will be good to include also an insert with schematic drawing of the BaF2 calorimeter.
(7) On page 4, the sentence on lines 88-90 is not understandable. What means “the used fit function is the sum of a Crystal ball, used to fit the peak, with …”. What is Crystal ball?
(8) The measurement and simulated data included in Table 2 are confusing. The reported “widths”, I suppose these are FWHM of the peaks, scatter randomly as function of energy, both in the simulations and in the measurements. What is the reason for it?
(9) In Sect. 2.1.2, the BaF2 detector tests are described. In the last sentence it is stated that 7.72 MeV α line will be used to monitor the performance of the detector. Why this line, and not the whole spectrum? In the fit in Fig. 3, it is not clear how the background for the 7.72 MeV line is taken into account.
(10)In Sect. 3, the NRSS is described. The core of this device is the ensemble of LYSO and BaF2 detectors. A better description of this assembly is needed, e.g. what are the dimensions of the LYSO detector, how the BaF2 detectors are placed with respect with the LYSO detector and what are their dimensions?
(11)On page 5, line 138, is stated that “realistic beams”, with a reference to the paper of Cardarelli et al., NIM B 355, 237 (2015), which describes the collimation system and provides examples for 2.5 MeV and 5 MeV beams after the collimator. It is not clear from the text, what is the energy of the beam, which was used in the simulations. In Fig. 5, there is a label “3 MeV”, does it denote the energy of the gamma beam? Please, explain in the text.
(12)The lines used in Fig. 5 are confusing, since they use the same colour. The notations in the figure are not self-explaining. I suggest to explain them in the figure caption.
(13)On page 7, the paragraph in lines 174-183, starts with the statement that an analytical model has been developed to predict the GPI system response. However, the authors do not provide any description of the model, neither do they refer to any published work. The information, which is provided in Table 3, cannot be understood, since these numbers cannot be compared with anything.
(14)What is reason for the fluctuation between 4 and 8 MeV and the raise above 16 MeV for the expected beam-energy resolution?
(15)Fig. 9 shows a photograph of the low-energy calorimeter detector. I would suggest to provide a schematic drawing, instead, or provide a second section of the figure with a schematic drawing as in all other cases.
Author Response
Response provided in the attached file.

Round 2
Reviewer 1 Report
I was not aware that this is a conference proceedings. Still, some corrections are needed:
1) In the introduction proper reference must be given to work done elsewhere, cf. the comments by one of the other reviewers suggesting relevant other facilities in the US and Japan. The authors must absolutely find the space to cite at least the latest review of the most important other gamma beams worldwide. If need be, they have to sacrifice a table or a figure to make the needed space.
2) The response by the authors on my criticism of section 2.1.1 and table 2 seems to indicate that the error bars given in the table must be revised upwards, so that the inconsistencies that I remarked upon are no longer significant.
3) Figure 3, the authors convinced me that Po-210 is negligible, so the label should be amended: Either delete mention of this negligible line, or write it only in brackets and not before the more important Rn-222 line.
4) The discussion of Birks' law by the authors is not convincing me completely. When looking at the relevant discussion in the book by Birks, it seems that for MeV alpha energies, the light output grows faster than linearly - this should lead to an even stronger improvement in relative resolution than what is expected for linearity.
Reviewer 2 Report
Second Review Report:
A characterization system for the monitoring of ELI-NP gamma beam, by Rita Borgheresi et al. (part of the EuroGammaS proposal for the ELI-NP Gamma beam System), submitted to Universe.
This is the second time I received this paper for review. I also received the reply of the authors to my first review, which I find disturbing. I state from the outset that this reviewer will not indulge in the exchange of polemics and I ask the authors to refrain from using non-professional language such as “the best in the world” or “useless” (review).
As I already outlined in my review of the first version of the paper a detailed description of the proposed systems was missing in the first version of the paper and it is still missing in the second version of the paper. I request the authors read again my first review and reply using a time honored rebuttal with listing the exact place(s) in the paper where revision where made in response to my previous review.
In the intervening time I consulted a number of leading authorities in the field in order to get some feedback and grasp the “bigger picture”. As such I sent my first review, the reply of the authors and the second version of the paper with requests for feedback which I consider confidential. I received similar independent statements that I consider as reflecting the views of the community which are summarized below:
1. The authors do not seem to be in full command and indeed not in contact with the field of gamma-beam monitoring. The authors failed to include in their paper the most elementary review of the field and fail to even refer and compare their proposed system to other working systems. Such an elementary review of the literature (with comprehensive referrals) must be added to the paper to demonstrate full command of the field.
2. The current paper does not provide the necessary details to evaluate their proposed systems. For example, it is now clear that ELI-NP will have two systems to monitor the gamma beam: the monitoring systems proposed here and a second one to be constructed independently by the ELI-NP itself on the other side of the wall in the target area. In the very likely event of a disagreement between the measurements of the two independent systems, a very detailed description of the proposed systems is required to understand and evaluate the anticipated disagreements.
3. The lack of communication between scientists working on the two systems for monitoring the gamma-beams (that will be placed on either side of the wall at the ELI-NP) must be stopped. It is inconceivable that the community was kept in the dark on the proposed gamma-beam monitor systems which is discussed in this paper for the first time. In that sense it is now essential to have a paper that fully describes the gamma-monitoring systems and allow for comparison between measurements that will be carried out on both side of the wall at the ELI-NP. A paper discussing the proposed systems is necessary to facilitate a discussion between the scientists involved in the ELI-NP project.
4. The authors failed to seriously consider the effect of pileup due to the time structure of the gamma-beam. The authors must demonstrate with hard numbers and simulations that pileup will not render their system obsolete. It is essential for the authors to provide hard numbers of the pulse shapes (rise and fall times), and include a realistic estimate of pileup.
This reviewer for one (together with others) is not convinced the authors gave the issue of pileup the necessary considerations.
I note that the authors repeatedly stated in their reply that this is “only a paper for a conference proceeding”. I understand that the editors agreed to allow the authors to have more space to describe the system. This is a must. In fact, it is very likely that this paper will be the only paper that will be published on the proposed gamma-beam monitoring systems and as such if additional space is required, I recommend that it will be provided to the authors. Simply put, there are close to 1,000 users in many countries that depend on the delivery of gamma-beams at the ELI-NP. The gamma-beam system costs many tens of million dollars and these together require a better description of the systems.
Reviewer 3 Report
The authors have taken into account all but one of my critisism and suggestions for improvement of the text of their manuscript "A characterization system for monitoring of ELI-NP' . The one, which is not considered, is related to the fact that the introduction does not provide relevant references, addressing similar systems, which are already operational in other laboratories.
The authors respond to this criticism, that this is a conference paper and they have space limitations. I am sorry to say, but I do not accept such a statement. A scientific paper must reveal the status of the field, such that the reader should be able to judge how the work, which is presented, compares with the state-of-the-art. Therefore, I insist that the authors provide such information.
Related to space, I suggest, to describe the content of Table I with a text. In this way, it will take much less space. In lines 142-146 and 155-161, the authors use bullets. I suggest to use plain text, which will also save space.
In the conclusion, I do not find a statement whether the proposed devices will provide monitoring of all beam parameters at the level defined in Table I.
I suggest some minor language corrections:
line 8: the word 'peculiar' seems to me not proper;
line 73: a recoil gamma-ray
line 82: 0.2-0.4 (unit is missing)line: 102: low-end tail
line 133: replace gamma-particles with gamma-rays
line 152: ... emerges that there are two background sources...
line 172: I suggest to end sentence after 'MC simulations' and 'start a new sentence with 'In Fig. 5'
line 185: comma after 'Outside the chamber'
line 207: change 'gamma' with 'photons'. Instead of 'But the photon energy' use 'However, the photon energy...'
line 241: write 7 with word (numbers from 1 to 12 in text are written with words, numbers above 12 are with numbers)
line 267: take out space between 1 and %
This list does not pretend to cover all needed language corrections. Therefore, I ask the authors to check the text once more.
With these corrections done, I think the paper is worth been published.
